# Low 25(OH)D Level Is Associated with Severe Course and Poor Prognosis in COVID-19

**DOI:** 10.3390/nu13093021

**Published:** 2021-08-29

**Authors:** Tatiana L. Karonova, Alena T. Andreeva, Ksenia A. Golovatuk, Ekaterina S. Bykova, Anna V. Simanenkova, Maria A. Vashukova, William B. Grant, Evgeny V. Shlyakhto

**Affiliations:** 1Clinical Endocrinology Laboratory, Department of Endocrinology, Almazov National Medical Research Centre, 194021 Saint-Petersburg, Russia; arabicaa@gmail.com (A.T.A.); ksgolovatiuk@gmail.com (K.A.G.); bykova160718@gmail.com (E.S.B.); annasimanenkova@mail.ru (A.V.S.); e.shlyakhto@almazovcentre.ru (E.V.S.); 2Botkin Clinical Infectious Hospital, 195067 Saint-Petersburg, Russia; mavashukova@yahoo.com; 3Sunlight, Nutrition, and Health Research Center, P.O. Box 641603, San Francisco, CA 94164-1603, USA; williamgrant08@comcast.net

**Keywords:** vitamin D deficiency, 25(OH)D, obesity, COVID-19

## Abstract

We evaluated associations between serum 25-hydroxyvitamin D [25(OH)D] level and severity of new coronavirus infection (COVID-19) in hospitalized patients. We assessed serum 25(OH)D level in 133 patients aged 21–93 years. Twenty-five (19%) patients had severe disease, 108 patients (81%) had moderate disease, and 18 (14%) patients died. 25(OH)D level ranged from 3.0 to 97.0 ng/mL (median, 13.5 [25%; 75%, 9.6; 23.3] ng/mL). Vitamin D deficiency was diagnosed in 90 patients, including 37 with severe deficiency. In patients with severe course of disease, 25(OH)D level was lower (median, 9.7 [25%; 75%, 6.0; 14.9] ng/mL), and vitamin D deficiency was more common than in patients with moderate course (median, 14.6 [25%; 75%, 10.6; 24.4] ng/mL, *p* = 0.003). In patients who died, 25(OH)D was 9.6 [25%; 75%, 6.0; 11.5] ng/mL, compared with 14.8 [25%; 75%, 10.1; 24.3] ng/mL in discharged patients (*p* = 0.001). Severe vitamin D deficiency was associated with increased risk of COVID-19 severity and fatal outcome. The threshold for 25(OH)D level associated with increased risk of severe course was 11.7 ng/mL. Approximately the same 25(OH)D level, 10.9 ng/mL, was associated with increased risk of mortality. Thus, most COVID-19 patients have vitamin D deficiency; severe vitamin D deficiency is associated with increased risk of COVID-19 severity and fatal outcome.

## 1. Introduction

Several studies conducted since the onset of the COVID-19 pandemic have shown that vitamin D deficiency can increase the incidence and worsen the course of acute respiratory viral infection caused by SARS-CoV-2 [1,2]. The immunomodulatory effects of vitamin D are well studied and are associated largely with the expression of the CYP27B1 enzyme and the presence of the vitamin D receptor in immune system cells [3,4,5]. Through several mechanisms, vitamin D may reduce the risk of bacterial and viral infection by creating a barrier, involving adaptive and humoral immunity [6]. Vitamin D is a potent stimulator of the monocyte/macrophage responses to bacterial infection, and thus it is an important participant in the innate immune response [7]. By contrast, inducing antimicrobial peptides, cathelicidin LL-37 [8,9] and defensins [10], vitamin D enhances cellular immunity, increases the synthesis of the NF-κB inhibitor IκBα, and decreases expression of proinflammatory genes [11]. Vitamin D also modulates humoral immunity [6,12,13] by suppressing interleukin 2 (IL-2) and interferon production and stimulating type 2 T-helper cytokine’s production [12,13,14]. The optimal vitamin D status can promote immunoregulatory functions in conditions of viral respiratory infection and overall can influence the altered immune-inflammatory COVID-19 reactivity at least by down-regulating overly exuberant cytokine responses that comprise pathological cytokine storm [15].

At the same time, considering the role of vitamin D in the activity of the renin–angiotensin–aldosterone system, researchers believe that it controls the amount of mRNA and the expression of angiotensin-converting enzyme 2, which determines the protective function against various respiratory infections. Moreover, vitamin D can suppress DPP-4/CD26, the putative adhesion molecule for SARS-CoV-2 to enter the cell [16,17].

Analyzing data on risk factors for COVID-19, we note their similarity to factors contributing to vitamin D deficiency. Those factors include age, sex, and race [18,19,20,21]; seasonality of incidence of acute respiratory viral infection, which accounts for the lowest concentration of 25-hydroxyvitamin D [25(OH)D] [22,23]; presence of obesity and type 2 diabetes mellitus [24,25]; and smoking [26].

The vitamin D level in patients with COVID-19 can be judged based on the results of single studies. Positive PCR tests for COVID-19 were more common in individuals with lower 25(OH)D levels [27]. Previously, we published data concerning the high incidence of vitamin D deficiency in residents of the northwest region of the Russian Federation [28]. That analysis was a prerequisite for this study to assess serum 25(OH)D level in hospitalized COVID-19 patients.

Objective of study: To assess vitamin D status in patients with community-acquired viral pneumonia with a confirmed diagnosis of COVID-19 and to match 25(OH)D value with disease severity.

## 2. Materials and Methods

We analyzed records of 161 patients with a new COVID-19 infection, hospitalized between April and December 2020 at Botkin Clinical Infectious Hospital (St. Petersburg, Russia; latitude, 59° N). Demographic data, information on the clinical course and infection severity, presence of concomitant diseases and drug therapy, and results of computed tomography (CT) and laboratory examinations were collected. Alcohol abuse was an exclusion criterion.

Pneumonia was established by means of chest CT without intravenous contrast enhancement. Volume of lung tissue lesions was described as follows: CT-1, lesion volume <25%; CT-2, lesion volume 25–50%; CT-3, lesion volume 50–75%; CT-4, lesion volume >75%.

Serum 25(OH)D level was detected by chemiluminescence immunoassay on micro particles (Abbott Architect c8000, Chicago, IL, USA, intra-assay CV of 1.60–5.92%, inter-assay CV ranged from 2.15 to 2.63%). According to Russian and international guidelines [29,30], normal vitamin D status was considered to be 25(OH)D ≥30 ng/mL (≥75 nmol/L); for insufficiency, ≥20 and <30 ng/mL (≥50 and <75 nmol/L); for deficiency, <20 ng/mL (<50 nmol/L), and for severe vitamin D deficiency, less than 10 ng/mL (<25 nmol/L). The 25(OH)D level between 10 to 20 ng/mL was assessed as mild deficiency in this work. The reference interval for serum 25(OH)D level determination was 3.4–155.9 ng/mL.

We also checked plasma glucose level and inflammatory reaction markers: C-reactive protein (CRP), IL-6, and ferritin. All parameters were measured at the time of admission (baseline), and their maximal values (max) were fixed.

Blood glucose level was measured using an automatic biochemistry analyzer (Cobas Integra c311 [Roche Diagnostics GmbH, Mannheim, Germany]) and diagnostic kits. Serum IL-6 concentration was determined by enzyme-linked immunosorbent assay on Bio-Rad 680 Microplate Reader equipment (Hercules, CA, USA), using appropriate reagent kits (Vector-Best; Novosibirsk, Russia). Results were processed using Zemfira 4 software for ELISA Bio-Rad analyzer (reference range, 0–7 pg/mL). An automatic biochemistry analyzer (Cobas Integra 400 [Roche Diagnostics GmbH, Mannheim, Germany]) and corresponding diagnostic kits from that manufacturer were used to determine level of CRP by means of turbidimetric method (reference range, 0–5 mg/L). Ferritin level was measured on an Abbott Architect c8000 analyzer (Chicago, IL, USA; reference range, 64–111 μmol/L).

Statistical processing of research results was carried out using the Statistica v. 10 package (StatSoft; Tulsa, OK, USA), with the help of standard methods of variation statistics. Between-group comparison was carried out using the Mann–Whitney criteria for incorrect distribution; results are presented as median (Me) and interquartile range [25%; 75%], as well as mean (M) and standard deviation (SD) for Student criterion in correct distributed parameters. Associations between quantitative parameters were assessed using Spearman’s correlation coefficient. To describe relative risk, we calculated the odds ratio (OR), with 95% confidence interval (95% CI) calculated with Fisher’s exact method. We explored the association between 25(OH)D level and both COVID-19 severity and fatal outcome using logistic regression (adjusting for age and comorbidities), with results expressed as β coefficients and 95% CI. The criterion for the statistical reliability of the obtained results was *p* < 0.05.

## 3. Results

As mentioned, vitamin D status was detected in 161 patients. We excluded data from 10 pregnant women, 5 patients with confirmed human immunodeficiency virus infection, and 13 patients taking replacement renal therapy for stage 5 chronic kidney disease. Thus, the final analysis included data from 133 COVID-19 patients (76 men [57%] and 57 women [43%]), aged 21–93 years (mean, 52 ± 14 years).

Based on disease severity, patients were grouped into moderate and severe course. Most hospitalized patients (81%) had a moderate course of disease with CT-confirmed lung damage as CT-2 and CT-3. At the same time, among patients with severe course, 56% had lung damage as CT-4. Severe patients were older and more often had obesity, diabetes mellitus, and cardiovascular diseases, especially coronary artery disease (CAD; Table 1).

Both baseline and maximal serum CRP, IL-6, and ferritin levels, as well as maximal glucose level in patients with a severe course, were expectedly higher than in patients with a moderate course, characterizing a prominent immune-inflammatory response.

Only 12 patients (9%) had a normal vitamin D status, whereas 91% were insufficient (23%) or deficient (mild deficiency in 40% and severe in 28%). Serum 25(OH)D level in severe-course patients was significantly lower than in moderate-course patients. Moreover, the number of patients with severe vitamin D deficiency [serum 25(OH)D level less than 10 ng/mL] in the severe-course group was larger than that in the moderate-course group (Table 1).

The number of obese patients in the vitamin D deficiency and severe deficiency groups tendered to be larger than in the vitamin D insufficiency group and in the group with normal 25(OH)D level, though the finding had no statistical significance (Table 2). Moreover, prevalence of CAD and DM was significantly higher in patients with vitamin D deficiency and severe deficiency. As noted, vitamin D mild and severe deficiency was associated with severe course of COVID-19. Thus, 35% of severe vitamin D-deficient patients and 15% of mild vitamin D-deficient ones had a severe course of COVID-19.

Analyzing factors possibly predisposing to death in COVID-19 infection, we found that patients who died were older and significantly more often had obesity, arterial hypertension, or CAD. Moreover, patients who died expectedly had higher blood glucose, CRP, IL-6, and ferritin levels. Vitamin D mild and severe deficiency was strongly associated with death incidence (Table 3).

We observed an inverse relationship between serum 25(OH)D level and CRP max level (*R* = −0.21; *p* = 0.02) and ferritin max level (*R* = −0.24; *p* = 0.01). Serum 25(OH)D level also negatively correlated with glucose max level (*R* = −0.25; *p* = 0.04).

Moreover, a positive correlation existed between glucose max level and baseline inflammatory marker levels: CRP (*R* = 0.26; *p* = 0.003), ferritin (*R* = 0.18; *p* = 0.04), and IL-6 (*R* = 0.20; *p* = 0.04). The correlation was also present for their max values: CRP (*R* = 0.44; *p* = 0.001), ferritin (*R* = 0.22; *p* = 0.03), and IL-6 (*R* = 0.26; *p* = 0.006).

We did not find correlation between 25(OH)D level and CT data in this population.

The threshold for 25(OH)D level associated with increased risk of severe course in this population was 11.7 ng/mL (AUC_area_ = 0.69; sensitivity, 71%; and specificity, 68%; *p* = 0.003) (Figure 1a). Approximately the same 25(OH)D level was associated with increased risk of mortality: 10.9 ng/mL (AUC_area_ = 0.75; sensitivity, 74%; and specificity, 72%; *p* = 0.001) (Figure 1b). That level corresponds to vitamin D deficiency.

We evaluated a possible contribution of vitamin D status and other predictors, such as age, sex, and comorbidities, to the risk of severe course (Table 4) and death (Table 5) in COVID-19 using multivariate ordered logistic regression analysis with both the adjusted and unadjusted models. To adjust for confounding factors, we used two models: model 1 was adjusted for age and sex, whereas model 2 was adjusted for age, sex, and comorbidities. Serum 25(OH)D level < 10.0 ng/mL increased risk of severe coronavirus infection by 3.79 times (95% CI, 1.53–9.39; *p* = 0.004) and death by 4.07 times (95% CI, 1.46–11.35; *p* = 0.007). In the unadjusted model, obesity appeared to be a significant predictor for severity (OR = 6.57; 95% CI, 2.57–16.78; *p* = 0.000) (Table 4) and death (OR = 6.52; 95% CI, 2.23–19.02; *p* = 0.001) (Table 5), and it significantly correlated with 25(OH)D level (*r* = –0.18; *p* = 0.04). Therefore, obesity was not included in the adjusted model. Sex was not a significant risk factor, whereas age and major comorbidities had significant relationships with severity and mortality in the unadjusted models only. By contrast, age was an independent predictor only in model 1 for severity (OR = 1.04; 95% CI, 1.01–1.07; *p* = 0.04), whereas no significance was evident for model 2. For death, age was a strong independent predictor in both model 1 (OR = 1.09; 95% CI, 1.03–1.16; *p* = 0.002) and model 2 (OR = 1.07; 95% CI, 1.01–1.15; *p* = 0.03). Serum 25(OH)D levels < 10 ng/mL in model 1 (OR = 4.09; 95% CI, 1.58–10.67; *p* = 0.004) and model 2 (OR = 4.17; 95% CI, 1.54–11.27; *p* = 0.005) were strongly associated with COVID-19 severity. The same pattern concerning severe 25(OH)D deficiency was observed for mortality in model 1 (OR = 5.68; 95% CI, 1.74–18.52; *p* = 0.004) and in model 2 (OR = 5.79; 95% CI, 1.66–20.22; *p* = 0.006).

## 4. Discussion

The available data indicate that vitamin D therapy in people with vitamin D insufficiency and deficiency reduces the likelihood of developing acute respiratory viral infections by 42% [31] and that patients with vitamin D deficiency have a longer, more severe course of disease [32]. Analysis of 25(OH)D levels in COVID-19 patients in China showed a high incidence of vitamin D deficiency in winter and a possible relationship between low vitamin D supply and severity and outcomes of the disease [33,34]. The results of a recent study showed that patients with severe coronavirus infection have the lowest serum 25(OH)D levels [35], in accordance with the data obtained here. Thus, vitamin D deficiency and obesity were most often reported in fatal COVID-19 patients [36,37] The results of the present study show that obesity independently worsens disease prognosis by at least 6 times, requiring an integrated approach in managing such patients. Moreover, severe vitamin D deficiency appears to be a strong independent negative predictor, even when adjusted for age, sex, and comorbidities, that worsens disease prognosis even by 4 times. Though in this study DM did not increase the risk of either severity or mortality of COVID-19 patients, we found associations between serum 25(OH)D level and plasma glucose concentration on one hand and between inflammatory markers and glucose level on the other hand that confirms the relationship between these conditions, and corresponds with previous data [38].

Our results correspond to the data presented in systematic review and meta-analysis of 23 studies that summarized all the existing knowledge concerning the role of vitamin D in COVID-19. According to the presented data, vitamin D deficiency increases the chance of severe COVID-19 development for about five times (OR = 5.1, 95% CI, 2.6–10.3), while there was no significant association between vitamin D status and increased mortality rates (OR = 1.6, 95% CI, 0.5–4.4) [39].

A main factor contributing to COVID-19 severity is believed to be development of the cytokine storm, the uncontrolled release of various inflammatory markers (such as CRP, IL-6, ferritin, tumor necrosis factor α or neutrophil-to-lymphocyte ratio (NLR)). For example, intensive care unit COVID-19 patients have the highest concentrations of IL-1β, IL-6, and IL-6 to IL-10 ratio [40]. Additionally, increased NLRs and decreased eosinophil counts are typical for severe COVID-19 patients, while neutrophils and lymphocytes counts demonstrate respectively positive and negative correlation with COVID-19 severity [41]. Our data confirmed that IL-6, CRP and ferritin levels are higher in severe COVID-19 cases and in patients with fatal outcome and showed negative correlations between these parameters and 25(OH)D.

The cytokine storm can activate intravascular coagulation, forming the basis for multiorgan injury, which is mediated mainly by inflammatory cytokines such as IL-6 [42]. By contrast, coronavirus can directly affect endothelial cells, causing cell death, and induces a cytopathic effect on airway epithelial cells [43]. Additionally, SARS-CoV-2 can affect the alveolar cells by ACE2 binding and suppress surfactant production. This damage might be prevented by vitamin D, as in vitro and in vivo studies have shown that 1.25(OH)2-D induces type II pneumocyte proliferation and surfactant synthesis in the lungs. These data are confirmed in clinical studies by positive correlation between vitamin D status and lung tissue lesions volume according to CT evaluation [44,45]. On the other hand, we did not find similar interlinks in our work.

IL-6 itself can increase the severity of COVID-19 by up-regulating angiotensin-converting enzyme 2 receptor and inducing cathepsin L production in macrophages, thus mediating the cleavage of the S1 subunit of the coronavirus surface spike glycoprotein. The latter is necessary for coronavirus to enter human host cells and to cause all further reactions. Low vitamin D concentration is associated with high IL-6 production, whereas vitamin D supplementation has an anti-inflammatory effect [46].

Taking into account vitamin D’s immunomodulatory effects, particularly the inhibition of NF-κB by increasing synthesis of IκBα [47,48], we can assume that vitamin D intake and the consequent achievement of a 25(OH)D concentration of 40–50 ng/mL (100–125 nmol/L) might have a positive effect in patients with coronavirus respiratory infections such as Middle East respiratory syndrome, SARS-CoV, and SARS-CoV-2 [11,49]. This is supported by data from a few studies showing that use of large vitamin D doses in critically ill patients with viral and bacterial pneumonia, under mechanical ventilation, leads to shortening of intensive care unit treatment duration and prognosis improvement [50]. However, further research is required to obtain more reliable information on vitamin D’s role in preventing and treating new coronavirus infection.

There are several limitations in the study. First, it is a single-center study with a relatively small sample size, while a larger cohort of COVID-19 patients is preferable to better assess vitamin D status and severity/outcomes of the disease. Secondly, only a part of the immune markers was included in the analysis; we do not have NLRs data for this cohort, so the immune response to SARS-CoV-2 should be characterized in more detail in the future. There were no anthropometric data in patients’ medical histories for us to calculate BMI, despite higher BMI being known as a strong predictor for vitamin D deficiency as well as COVID-19 severity. Moreover, this study is not a prospective one and does not provide the information regarding the relationship between vitamin D status dynamic and immune response, as well as the outcomes of COVID-19.

## Figures and Tables

**Figure 1 nutrients-13-03021-f001:**
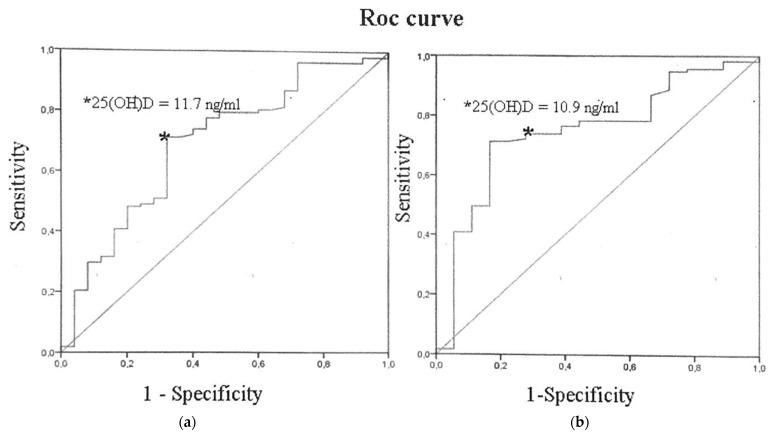
25-Hydroxyvitamin D level’s association with increased risk of (**a**) severe course, *p* = 0.003; (**b**) mortality, *p* = 0.001.

**Table 1 nutrients-13-03021-t001:** Patients’ characteristics in relation to COVID-19 severity.

Parameter	Severe Course *n* = 25	Moderate Course *n* = 108	*p*
Age, y, M ± SD	57 ± 3	51 ± 1	**0.02**
Sex, m/f, *n* (%)	15(60)/10(40)	61(57)/47(44)	0.75
Obesity, *n* (%)	16 (64)	23 (21)	**0.00**
AH, *n* (%)	15 (60)	46 (43)	0.12
CAD, *n* (%)	11 (44)	25 (23)	**0.04**
DM, *n* (%)	8 (32)	18 (17)	**0.00**
Death, *n* (%)	15 (60)	3 (3)	**0.00**
Volume of lung tissue lesions (CT), *n* (%)			**0.00**
0	0	7 (7)	
1	1 (4)	19 (18)	
2	5 (20)	41 (38)	
3	5 (20)	32 (30)	
4	14 (56)	9 (8)	
25(OH)D, ng/mL, Me [25; 75]	9.7 [6.0; 14.9]	14.6 [10.6; 24.4]	**0.00**
Vitamin D status, *n* (%)			**0.003**
Normal	1 (4)	11 (10)	
Insufficiency	3 (12)	28 (26)	
Mild deficiency	8 (32)	45 (42)	
Severe deficiency	13 (52)	24 (22)	
Bed days, M ± SD	21.0 ± 2.5	17.0 ± 0.9	0.16
Glucose max, mmol/L, Me [25%; 75%]	10.3 [8.4; 18.4]	6.15, 0; 9.7	**0.00**
CRP baseline, mg/L, Me [25%; 75%]	64.7 [36.4; 200.0]	34.7 [15.9; 89.6]	**0.01**
CRP max, mg/L, Me [25%; 75%]	265.1 [182.2; 322.0]	60.0 [21.4; 137.3]	**0.00**
IL-6 baseline, pg/mL, Me [25%; 75%]	22.0 [10.8; 75.0]	7.8 [2.4; 20.7]	**0.001**
IL-6 max, pg/mL, Me [25%; 75%]	36.4 [20.1; 282.0]	10.4 [2.8; 25.1]	**0.00**
Ferritin baseline, μg/L, Me [25%; 75%]	895.4 [317.1; 1581.7]	357.3 [172.2; 811.3]	**0.01**
Ferritin max, μg/L, Me [25%; 75%]	1347.6 [835.6; 2197.1]	496.1 [257.4; 1057.4]	**0.00**

M, mean; SD, standard deviation; m, men; f, women; AH, arterial hypertension; CAD, coronary artery disease; DM, diabetes mellitus; CT, computed tomography; CRP, C-reactive protein; IL-6, interleukin 6; *p* < 0.05 values are bolded.

**Table 2 nutrients-13-03021-t002:** Vitamin D status in 133 COVID-19 patients.

Parameter	Deficiency *n* =90		Insufficiency *n* = 31	Normal *n* = 12	*p*
	Severe Deficiency *n* =37	Mild Deficiency *n* = 53			
Age, y, M ± SD	52 ± 3	53 ± 2	49 ± 2	51 ± 3	0.39 * 0.74
Sex, m/f, *n* (%)	27 (73)/ 10 (27)	26 (49)/ 27 (51)	14 (45)/ 17 (55)	9 (75)/ 3 (25)	0.17 *** 0.02**
Obesity, *n* (%)	13 (35)	19 (36)	6 (19)	1 (8)	0.06 * 0.36
AH, *n* (%)	19 (51)	28 (53)	11 (36)	3 (25)	0.09 * 0.43
CAD, *n* (%)	15 (41)	14 (26)	6 (19)	1 (8)	0.12 *** 0.03**
DM, *n* (%)	7 (19)	14 (26)	5 (16)	0	**0.00** *** 0.00**
Severe course, *n* (%)	13 (35)	8 (15)	3 (10)	1 (8)	0.15 *** 0.003**

*, compared with severe deficiency. M, mean; SD, standard deviation; m, men; f, women; AH, arterial hypertension; CAD, coronary artery disease; DM, diabetes mellitus; *p* < 0.05 values are bolded.

**Table 3 nutrients-13-03021-t003:** COVID-19 patient characteristics in relation to disease outcome.

Parameter	Death *n* = 18	Discharged *n* = 115	*p*
Age, y, M ± SD	62 ± 3	50 ± 1	**0.00**
Sex, m/f, *n* (%)	10 (56)/ 8 (44)	66 (57)/ 49 (43)	0.88
Obesity, *n* (%)	12 (67)	27 (24)	**0.00**
AH, *n* (%)	14 (78)	47 (41)	**0.004**
CAD, *n* (%)	11 (61)	25 (22)	**0.00**
DM, *n* (%)	6 (33)	20 (17)	**0.00**
Severe course, *n* (%)	15 (83)	10 (9)	**0.00**
Volume of lung tissue lesions (CT), *n* (%)			**0.00**
0	0	7 (6)	
1	0	20 (17)	
2	4 (22)	42 (37)	
3	3 (17)	34 (30)	
4	11 (61)	12 (10)	
25(OH)D, ng/mL, Me [25%; 75%]	9.6 [6.0; 11.5]	14.8 [10.1; 24.3]	**0.001**
Vitamin D status, *n* (%)			
Normal, *n* (%)	1 (6)	11 (10)	**0.02**
Insufficiency, *n* (%)	0	31 (27)	
Mild deficiency, *n* (%)	7 (39)	46 (40)	
Severe deficiency, *n* (%)	10 (55)	27 (24)	**0.005**
Bed days, M ± SD	16 ± 3	18 ± 1	0.27
Glucose baseline, mmol/L	7.0 [6.1; 10.0]	5.9 [5.0; 7.5]	**0.03**
Glucose max, mmol/L	10.8 [7.7; 18.4]	6.3 [5.0; 10.0]	**0.00**
CRP baseline, mg/L	74.6 [36.4; 168.2]	35.5 [16.3; 90.0]	**0.02**
CRP max, mg/L	255.5 [182.2; 308.0]	67.4 [22.2; 140.1]	**0.00**
IL-6 baseline, pg/mL	37.4 [10.8; 87.6]	8.3 [2.4; 20.7]	**0.00**
IL-6 max, pg/mL	37.7 [12.7; 453.5]	11.8 [2.9; 27.6]	**0.001**
Ferritin baseline, μg/L, *n* = 131	965.0 [680.1; 1581.7]	366.5 [172.2; 895.4]	**0.005**
Ferritin max, μg/L	1699.1 [1119.5; 2197.1]	536.1 [260.0; 1051.0]	**0.00**

M, mean; SD, standard deviation; m, men; f, women; AH, arterial hypertension; CAD, coronary artery disease; DM, diabetes mellitus; CT, computed tomography; Me, median, CRP, C-reactive protein; IL-6, interleukin 6; *p* < 0.05 values are bolded.

**Table 4 nutrients-13-03021-t004:** Predictors for COVID-19 severity (logistic regression analysis).

Predictor	Unadjusted		Model 1 Adjusted		Model 2 Adjusted	
	OR (95% CI)	*p*	OR (95% CI)	*p*	OR (95% CI)	*p*
Age	1.03 (1.01–1.07)	**0.04**	1.04 (1.01–1.07)	**0.04**	1.03 (0.99–1.08)	0.17
Male	1.16 (0.48–2.80)	0.75	0.77 (0.20–2.02)	0.59	0.85 (0.32–2.29)	0.75
Obesity	6.57 (2.57–16.78)	**0.000**				
AH	2.02 (0.83–4.91)	0.12			0.98 (0.28–3.43)	0.98
CAD	2.61 (1.05–6.46)	**0.04**			1.18 (0.34–4.08)	0.79
DM	2.35 (0.88–6.28)	0.08			2.25 (0.77–6.57)	0.14
25(OH)D < 20 ng/mL	2.97 (0.95–9.27)	0.06	2.72 (0.86–8.59]	0.09	2.48 (0.77–7.99)	0.13
25(OH)D < 10 ng/mL	3.79 (1.53–9.39)	**0.004**	4.09 (1.58–10.67)	**0.004**	4.17 (1.54–11.27)	**0.005**

Model 1 is adjusted for age and sex. Model 2 is adjusted for age, sex, and comorbidities. Data are presented as frequencies (%) and OR (95% CI). OR, odds ratio; CI, confidence interval; AH, arterial hypertension; CAD, coronary artery disease; DM, diabetes mellitus; *p* < 0.05 values are bolded.

**Table 5 nutrients-13-03021-t005:** Predictors for COVID-19 fatal outcome (logistic regression analysis).

Predictor	Unadjusted		Model 1 Adjusted		Model 2 Adjusted	
	OR (95% CI)	*p*	OR (95% CI)	*p*	OR (95% CI)	*p*
Age	1.07 (1.03–1.13)	**0.001**	1.09 (1.03–1.16)	**0.002**	1.07 (1.01–1.15)	**0.03**
Male	0.93 (0.34–2.52)	0.88	0.43 (0.13–1.41)	0.16	0.51 (0.15–1.69)	0.27
Obesity	6.52 (2.23–19.02)	**0.001**				
AH	5.06 (1.57–16.34)	0.07			1.59 (0.35–5.54)	0.56
CAD	5.66 (1.99–16.10)	**0.001**			1.40 (0.35–5.54)	0.63
DM	2.38 (0.79–7.08)	0.12			1.98 (0.57–6.92)	0.28
25(OH)D < 20 ng/mL	9.78 (1.26–76.14)	**0.03**	8.60 (1.07–69.12)	0.05	7.87 (0.96–64.43)	0.06
25(OH)D < 10 ng/mL	4.07 (1.46–11.35)	**0.007**	5.68 (1.74–18.52)	**0.004**	5.79 (1.66–20.22)	**0.006**

Model 1 is adjusted for age and sex. Model 2 is adjusted for age, sex, and comorbidities. Data are presented as frequencies (%) and odds ratio (95% CI). OR, odds ratio; CI, confidence interval; AH, arterial hypertension; CAD, coronary artery disease; DM, diabetes mellitus; *p* < 0.05 values are bolded.

## Data Availability

The data generated and analyzed during this study are included in this published article and its supplementary information files. Additional information is available from the corresponding author on reasonable request.

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
