# Peer review of "Low 25(OH)D Level Is Associated with Severe Course and Poor Prognosis in COVID-19"

_nutrients, 2021, doi:10.3390/nu13093021_

Round 1

Reviewer 1 Report

The correlation of serum vitamin D levels and course of COVID-19 is a hot topic in the recent international literature for the potential implications in the treatment of the disease.

The paper is of interest but I have several observations on it.

Data from patients with vitamin D deficiency are presented in a confusing manner (see Results, Table 2 and Table 3). It’s deeply incorrect to group under the term “deficiency” both deficient and severe deficient patients. As correctly stated in Materials and Methods section, the term “deficiency” can refer only to the group of patients with serum 25(OH)D level between 10 and 20 ng/ml and it cannot be used to include also “severe deficient” patients. After a recalculation (please, confirm its accuracy), data must be presented in the following way: severe deficient n = 37 (28%), deficient n= 53 (40%), insufficient n=31 (23%). Please correct deficient patients n = 90 (table 2) and 68% (second line page 4). As a consequence, correct the data of patients with vitamin D deficiency in all the sections of your work in appropriate manner.

  1. Please, read and report in the introduction as reference the following pertinent Editorial: Cutolo M, Paolino S, Smith V. Evidences for a protective role of vitaminD in COVID-19. RMD Open. 2020 Dec;6(3):e001454. doi: 10.1136/rmdopen-2020-001454.
  2. Please align data of “volume of lung tissue lesions” in table 1 correctly. Moreover, Data of table 4 and 5 are difficult to read. Please make appropriate changes.
  3. In the discussion, CRP and ferritin are reported as inflammatory cytokines. However, they are only inflammation markers. Please, correct properly.
  4. Discussion section should be implemented. As suggestions:
    1. Compare vitamin D serum levels of COVID-19 patients with general population of your Country, referring to your previous work “Karonova T, Andreeva A, Nikitina I, Belyaeva O, Mokhova E, Galkina O, Vasilyeva E, Grineva E. Prevalence of Vitamin D deficiency in the North-West region of Russia: A cross-sectional study. J Steroid Biochem Mol Biol. 2016 Nov;164:230-234. doi: 10.1016/j.jsbmb.2016.03.026. Epub 2016 Mar 21. PMID: 27013017” and to larger systematic reviews and/or metanalysis (for example “Dramé M, Cofais C, Hentzien M, Proye E, Coulibaly PS, Demoustier-Tampère D, Destailleur MH, Lotin M, Cantagrit E, Cebille A, Desprez A, Blondiau F, Kanagaratnam L, Godaert L. Relation between Vitamin D and COVID-19 in Aged People: A Systematic Review. 2021 Apr 17;13(4):1339. doi: 10.3390/nu13041339. PMID: 33920639; PMCID: PMC8073922);
    2. COVID-19 induces a pro-coagulative state, with frequent thrombotic events. Do you have any data regarding deep venous thrombosis or lung thromboembolism in your cohort patients? Moreover, a link between vitamin D and coagulation has been proposed and can be discussed. Only for example, you can refer to Zhang J, McCullough PA, Tecson KM. Vitamin D deficiency in association with endothelial dysfunction: Implications for patients with COVID-19. Rev Cardiovasc Med. 2020 Sep 30;21(3):339-344. doi: 10.31083/j.rcm.2020.03.131. PMID: 33070539”;
    3. Do you find any correlation between vitamin D status and CT evaluation of volume of lung tissue lesions? As reference to be reported in the literature “Sulli A, Gotelli E, Casabella A, Paolino S, Pizzorni C, Alessandri E, Grosso M, Ferone D, Smith V, Cutolo M. Vitamin D and Lung Outcomes in Elderly COVID-19 Patients. Nutrients. 2021 Feb 24;13(3):717. doi: 10.3390/nu13030717. PMID: 33668240; PMCID: PMC7996150” and “Fakhoury HMA, Kvietys PR, Shakir I, Shams H, Grant WB, Alkattan K. Lung-Centric Inflammation of COVID-19: Potential Modulation by Vitamin D. Nutrients. 2021 Jun 28;13(7):2216. doi: 10.3390/nu13072216. PMID: 34203190; PMCID: PMC8308422”.
  5. At last, several English changes are required. Of note:
    1. Page 1 - “affecting adaptive and humoral immunity”: please change “affecting” into “involving”.
    2. Page 1 - “one function of vitamin D is associated with the recognition of pathogenic microorganisms by macrophages”: please better explain this sentence.
    3. Page 1 - “by contrast” seems incorrect: please change it.
    4. Page 2, first line - “taking into account vitamin D’s known participation in activity of the renin-angiotensin-aldosterone system”.
    5. Page 3, first line of Results - “primarily vitamin D status was detected in 161 patients”: maybe you meant “vitamin D was first detected in 161 patients”.
    6. Page 4: “severe deficiency groups tendered to be larger”.
    7. Page 5: “the correlation also was present”: change into “the correlation was also present”.

Author Response

Dear Reviewer!

We would like to thank you for careful consideration of the manuscript entitled “Low 25(OH)D level is associated with severe course and poor prognosis in COVID-19” intended for publication in the Nutrients as an original research article.

Please find the revised manuscript and our responses to the comments. The changes made to the text have been highlighted.

Comments and Suggestions for Authors

The correlation of serum vitamin D levels and course of COVID-19 is a hot topic in the recent international literature for the potential implications in the treatment of the disease. The paper is of interest but I have several observations on it.

  • Data from patients with vitamin D deficiency are presented in a confusing manner (see Results, Table 2 and Table 3). It’s deeply incorrect to group under the term “deficiency” both deficient and severe deficient patients. As correctly stated in Materials and Methods section, the term “deficiency” can refer only to the group of patients with serum 25(OH)D level between 10 and 20 ng/ml and it cannot be used to include also “severe deficient” patients. After a recalculation (please, confirm its accuracy), data must be presented in the following way: severe deficient n = 37 (28%), deficient n= 53 (40%), insufficient n=31 (23%). Please correct deficient patients n = 90 (table 2) and 68% (second line page 4). As a consequence, correct the data of patients with vitamin D deficiency in all the sections of your work in appropriate manner.

Answer: We have corrected the data of patients with vitamin D deficiency in tables and other sections of our work. The 25(OH)D level between 10 to 20 ng/ml was assessed as mild deficiency in this work.

  • Please, read and report in the introduction as reference the following pertinent Editorial: Cutolo M, Paolino S, Smith V. Evidences for a protective role of vitamin D in COVID-19. RMD Open. 2020 Dec;6(3):e001454. doi: 10.1136/rmdopen-2020-001454.

Answer: We have reported the work of Cutolo M. et al in our introduction section.

  • Please align data of “volume of lung tissue lesions” in table 1 correctly. Moreover, Data of table 4 and 5 are difficult to read. Please make appropriate changes.

Answer: We have corrected the tables format and made the appropriate changes.

  • In the discussion, CRP and ferritin are reported as inflammatory cytokines. However, they are only inflammation markers. Please, correct properly.

Answer: The corrections have been done.  

  • Discussion section should be implemented. As suggestions:Compare vitamin D serum levels of COVID-19 patients with general population of your Country, referring to your previous work “Karonova T, Andreeva A, Nikitina I, Belyaeva O, Mokhova E, Galkina O, Vasilyeva E, Grineva E. Prevalence of Vitamin D deficiency in the North-West region of Russia: A cross-sectional study. J Steroid Biochem Mol Biol. 2016 Nov;164:230-234. doi: 10.1016/j.jsbmb.2016.03.026. Epub 2016 Mar 21. PMID: 27013017” and to larger systematic reviews and/or metanalysis (for example “Dramé M, Cofais C, Hentzien M, Proye E, Coulibaly PS, Demoustier-Tampère D, Destailleur MH, Lotin M, Cantagrit E, Cebille A, Desprez A, Blondiau F, Kanagaratnam L, Godaert L. Relation between Vitamin D and COVID-19 in Aged People: A Systematic Review. 2021 Apr 17;13(4):1339. doi: 10.3390/nu13041339. PMID: 33920639; PMCID: PMC8073922);

COVID-19 induces a pro-coagulative state, with frequent thrombotic events. Do you have any data regarding deep venous thrombosis or lung thromboembolism in your cohort patients? Moreover, a link between vitamin D and coagulation has been proposed and can be discussed. Only for example, you can refer to Zhang J, McCullough PA, Tecson KM. Vitamin D deficiency in association with endothelial dysfunction: Implications for patients with COVID-19. Rev Cardiovasc Med. 2020 Sep 30;21(3):339-344. doi: 10.31083/j.rcm.2020.03.131. PMID: 33070539”;

Answer: Thank you for having attached these articles, we have included the work of Zhang J et al in our discussion section.

  • Do you find any correlation between vitamin D status and CT evaluation of volume of lung tissue lesions? As reference to be reported in the literature “Sulli A, Gotelli E, Casabella A, Paolino S, Pizzorni C, Alessandri E, Grosso M, Ferone D, Smith V, Cutolo M. Vitamin D and Lung Outcomes in Elderly COVID-19 Patients. Nutrients. 2021 Feb 24;13(3):717. doi: 10.3390/nu13030717. PMID: 33668240; PMCID: PMC7996150” and “Fakhoury HMA, Kvietys PR, Shakir I, Shams H, Grant WB, Alkattan K. Lung-Centric Inflammation of COVID-19: Potential Modulation by Vitamin D. Nutrients. 2021 Jun 28;13(7):2216. doi: 10.3390/nu13072216. PMID: 34203190; PMCID: PMC8308422”.

Answer: We have not found the correlation between 25(OH)D level and CT data in this population. We have included these data to our results and discussion sections.

  • At last, several English changes are required. Of note:

Page 1 - “affecting adaptive and humoral immunity”: please change “affecting” into “involving”.

Page 1 - “one function of vitamin D is associated with the recognition of pathogenic microorganisms by macrophages”: please better explain this sentence.

Page 1 - “by contrast” seems incorrect: please change it.

Page 2, first line - “taking into account vitamin D’s known participation in activity of the renin-angiotensin-aldosterone system”.

Page 3, first line of Results - “primarily vitamin D status was detected in 161 patients”: maybe you meant “vitamin D was first detected in 161 patients”.

Page 4: “severe deficiency groups tendered to be larger”.

Page 5: “the correlation also was present”: change into “the correlation was also present”.

Answer: Thank you. The corrections have been done. 

Reviewer 2 Report

This is a retrospective cohort study aimed to evaluate the associations between low level of vitamin D and severity of COVID infection.

There are some inconsistencies/questions that should  be clarified:

1) Number of Patients:

-the authors analyzed 205 patients, were all these 205 patients “consecutive patients”?

For the measure of vitamin D the authors enrolled 133 patients (as well reported in table 1), but in table 2 are reported 170 patients. The numbers are not clear: how many patients (or how many were excluded) not have  available data for vitamin D, chest CT and other inflammatory data? The exact number of patients should be specified better.

2) For the inflammation, other hematological parameters (e.g  NLR or others) could be suitable to include in this work. The neutrophil-to-lymphocyte ratio (NLR) is common used as marker of systemic inflammation (ref. Liu X, Shen Y, Wang H, Ge Q, Fei A, Pan S. Prognostic significance of neutrophil-to-lymphocyte ratio in patients with sepsis: a prospective observational study. Mediators Inflamm. 2016;2016: 8191254), and NLR was also reported in recent papers regarding inflammation and covid

3) Obesity: For obesity, the value of BMI should be reported

4) Discussion needs to be written better. It is incomplete, too short and not always appropriate to explain the results. The authors should describe in text “relationship between 25(OH)D and xxx”, which just summarizes what is in the tables, e.g:

a) relationship between cytokine storm (IL-6), vitamin D and COVID

b) relationship about glucose levels and vitamin D and COVID, or ,obesity, diabetes, inflammation, vitamin D and COVID

c) About the vitamin D intake, it should be only mentioned shortly as future perspective.

4) REFERENCES:

At ref 1 more similar and more recent papers should be added:

- a) Vitamin D deficiency and the COVID-19 pandemic.Zemb P, Bergman P, Camargo CA Jr, Cavalier E, Cormier C, Courbebaisse M, Hollis B, Joulia F, Minisola S, Pilz S, Pludowski P, Schmitt F, Zdrenghea M, Souberbielle JC.J Glob Antimicrob Resist. 2020 Sep;22:133-134. doi: 10.1016/j.jgar.2020.05.006. Epub 2020 May 29.PMID: 32474141.

- b) Low Vitamin D Status at Admission as a Risk Factor for Poor Survival in Hospitalized Patients With COVID-19: An Italian Retrospective Study.Infante M, Buoso A, Pieri M, Lupisella S, Nuccetelli M, Bernardini S, Fabbri A, Iannetta M, Andreoni M, Colizzi V, Morello M.J Am Coll Nutr. 2021 Feb 18:1-16. doi: 10.1080/07315724.2021.1877580.

c) Evidence Regarding Vitamin D and Risk of COVID-19 and Its Severity.Mercola J, Grant WB, Wagner CL.Nutrients. 2020 Oct 31;12(11):E3361. doi:10.3390/nu12113361.

Author Response

Dear Reviewer!

We would like to thank you for constructive and helpful comments that you have kindly made on the manuscript entitled “Low 25(OH)D level is associated with severe course and poor prognosis in COVID-19” intended for publication in the Nutrients as an original research article.

Please find the revised manuscript and our responses to the comments. The changes made to the text have been highlighted.

Comments and Suggestions for Authors

This is a retrospective cohort study aimed to evaluate the associations between low level of vitamin D and severity of COVID infection.

There are some inconsistencies/questions that should be clarified:

1) Number of Patients:

-the authors analyzed 205 patients, were all these 205 patients “consecutive patients”?

For the measure of vitamin D the authors enrolled 133 patients (as well reported in table 1), but in table 2 are reported 170 patients. The numbers are not clear: how many patients (or how many were excluded) not have  available data for vitamin D, chest CT and other inflammatory data? The exact number of patients should be specified better.

Answer: We have analyzed data from 205 patients with a new COVID-19 infection in one Center. These were consecutive patients. Serum 25(OH)D level was detected only in 161 patients out of these 205 patients. We excluded data from 10 pregnant women, 5 patients with confirmed human immunodeficiency virus infection, and 13 patients taking replacement renal therapy for stage 5 chronic kidney disease (in total, 28 patients were excluded). Thus, the final analysis included data from 133 COVID-19 patients. So, other data such as chest CT and inflammatory parameters were analyzed for 133 COVID-19 patients.

 Such number as 170 patients is not completely correct. Such a number can be suspected while calculating severe deficient patients separately, while we included them into the total number of deficient patients. Now we have made some corrections to minimize such misunderstandings.  One of them, we correct the initial number of patients to 161. Thank you so much for careful examination and correcting the provided data which hopefully allowed us to make the presented material more comprehensible.

2) For the inflammation, other hematological parameters (e.g  NLR or others) could be suitable to include in this work. The neutrophil-to-lymphocyte ratio (NLR) is common used as marker of systemic inflammation (ref. Liu X, Shen Y, Wang H, Ge Q, Fei A, Pan S. Prognostic significance of neutrophil-to-lymphocyte ratio in patients with sepsis: a prospective observational study. Mediators Inflamm. 2016;2016: 8191254), and NLR was also reported in recent papers regarding inflammation and covid

Answer: It was one of limitations in our study: we do not have a NLR data for this population. We have mentioned this point in the discussion section and also have added a section “study limitations”.

3) Obesity: For obesity, the value of BMI should be reported

Answer: Also, we do not have anthropometric data of our patients to calculate the BMI, we only know about the presence of obesity as a nosological diagnosis.

4) Discussion needs to be written better. It is incomplete, too short and not always appropriate to explain the results. The authors should describe in text “relationship between 25(OH)D and xxx”, which just summarizes what is in the tables, e.g:

  1. a) relationship between cytokine storm (IL-6), vitamin D and COVID
  2. b) relationship about glucose levels and vitamin D and COVID, or ,obesity, diabetes, inflammation, vitamin D and COVID
  3. c) About the vitamin D intake, it should be only mentioned shortly as future perspective.

Answer: We have done these corrections in discussion section.

4) REFERENCES:

At ref 1 more similar and more recent papers should be added:

- a) Vitamin D deficiency and the COVID-19 pandemic. Zemb P, Bergman P, Camargo CA Jr, Cavalier E, Cormier C, Courbebaisse M, Hollis B, Joulia F, Minisola S, Pilz S, Pludowski P, Schmitt F, Zdrenghea M, Souberbielle JC.J Glob Antimicrob Resist. 2020 Sep;22:133-134. doi: 10.1016/j.jgar.2020.05.006. Epub 2020 May 29.PMID: 32474141.

- b) Low Vitamin D Status at Admission as a Risk Factor for Poor Survival in Hospitalized Patients With COVID-19: An Italian Retrospective Study.Infante M, Buoso A, Pieri M, Lupisella S, Nuccetelli M, Bernardini S, Fabbri A, Iannetta M, Andreoni M, Colizzi V, Morello M.J Am Coll Nutr. 2021 Feb 18:1-16. doi: 10.1080/07315724.2021.1877580.

  1. c) Evidence Regarding Vitamin D and Risk of COVID-19 and Its Severity.Mercola J, Grant WB, Wagner CL.Nutrients. 2020 Oct 31;12(11):E3361. doi:10.3390/nu12113361.

Answer: Thank you for having kindly attached these articles. We had the Mercola J et al results in our work. We have mentioned another works in the article and references.

Round 2

Reviewer 1 Report

Now can be published

Author Response

Dear Reviewer, thank you for help and advices